# Tuft Cells: Detectors, Amplifiers, Effectors and Targets in Parasite Infection

**DOI:** 10.3390/cells12202477

**Published:** 2023-10-18

**Authors:** Marta Campillo Poveda, Collette Britton, Eileen Devaney, Tom N. McNeilly, François Gerbe, Philippe Jay, Rick M. Maizels

**Affiliations:** 1Wellcome Centre for Integrative Parasitology, School of Infection and Immunity, University of Glasgow, Glasgow G12 8TA, UK; marta.campillopoveda@glasgow.ac.uk; 2School of Biodiversity, One Health and Veterinary Medicine, University of Glasgow, Glasgow G61 1QH, UK; collette.britton@glasgow.ac.uk (C.B.); eileen.devaney@glasgow.ac.uk (E.D.); 3Disease Control Department, Moredun Research Institute, Penicuik EH26 0PZ, UK; tom.mcneilly@moredun.ac.uk; 4Institute of Functional Genomics (IGF), University of Montpellier, CNRS, INSERM, F-34094 Montpellier, France; francois.gerbe@igf.cnrs.fr (F.G.); philippe.jay@igf.cnrs.fr (P.J.)

**Keywords:** cytokines, helminths, innate immunity, mucosal immunity, type 2 immunity

## Abstract

Tuft cells have recently emerged as the focus of intense interest following the discovery of their chemosensory role in the intestinal tract, and their ability to activate Type 2 immune responses to helminth parasites. Moreover, they populate a wide range of mucosal tissues and are intimately connected to immune and neuronal cells, either directly or through the release of pharmacologically active mediators. They are now recognised to fulfil both homeostatic roles, in metabolism and tissue integrity, as well as acting as the first sensors of parasite infection, immunity to which is lost in their absence. In this review we focus primarily on the importance of tuft cells in the intestinal niche, but also link to their more generalised physiological role and discuss their potential as targets for the treatment of gastrointestinal disorders.

## 1. Introduction

Tuft cells are specialized epithelial cells that are distributed across many barrier surfaces, but are particularly involved in the detection, amplification, and effector functions of the immune response to parasite infections in the gut [1,2,3]. In the case of intestinal helminth (worm) parasites, tuft cells detect and respond to their presence by releasing the alarmin interleukin-25 (IL-25), which activates group 2 innate lymphoid cells (ILC2s) to initiate the anti-helminth immune response through type 2 helper T cells (Th2) [4,5,6]. Hence, tuft cells are a critical component of the pathway through which innate immunity triggers and expands the adaptive immune system [7].

Th2 activation leads to the production of type 2 cytokines, such as interleukin-4 (IL-4), interleukin-5 (IL-5), and interleukin-13 (IL-13), which promote the expulsion of parasites from the gut [8]. Notably, IL-4 and IL-13 act on intestinal stem cells to promote differentiation of secretory goblet and tuft cells which expand in numbers during helminth infection. This expansion allows intestinal tuft cells to serve as effectors by secreting small pharmacologically active molecules, including leukotrienes, prostaglandins, and acetylcholine [9].

Therefore, tuft cells are important players in the host–parasite interaction in the gut and represent potential targets for the development of novel therapies against parasitic infections (Figure 1). For example, priming animals to expand tuft cell activity or number prior to parasite exposure, with molecules such as succinate, might be an effective strategy in the control of gastrointestinal helminths, particularly for livestock in which helminth infections are a global problem due to widespread and increasing resistance to current anthelmintic drugs [10].

Tuft cells in the different barrier tissues show important anatomical and physiological differences; for example, in the airways they are directly innervated, which may facilitate an immediate response to the entry of noxious substances. In the gut, where all epithelial cell types are replaced with rapidity, nervous system interactions may be mediated indirectly (for example by ILC2 neuromedin). Further heterogeneity is seen within the intestinal tuft cell populations, with proximal–distal gradients in receptor expression (in the case of succinate receptor) or a dichotomy between CD45– and CD45+ tuft cells that has been proposed to demarcate “Tuft-1” and “Tuft-2” cells, associated with a more neuronal or lymphoid gene expression pattern, respectively [11]. 

In this review, we focus primarily on tuft cells in the context of gastrointestinal helminth infections. It should also be noted that dysregulation of tuft cells has been implicated in other gastrointestinal disorders [12,13], including colorectal cancer, in which tuft cell genes are highly expressed [14], while in both radiation injury [15] and inflammatory bowel disease models [16], tuft cells fulfil a pro-regenerative role. Within the gastrointestinal tract, it is notable that tuft cells also populate the bile duct, where they restrain inflammatory responses in a bile acid-sensitive manner [17]. While the mechanisms linking tuft cells with cancer and intestinal inflammation are poorly characterised, understanding the biology and functions of these cells in the gut may provide novel insights into the pathogenesis and treatment of a range of gastrointestinal diseases.

## 2. Tuft Cell Differentiation and Gene Expression

In pathogen-free laboratory animals, intestinal tuft cells are found at a low frequency, but rapidly increase in number in response to molecular cues or luminal signals such as pathogen colonization as a result of preferential differentiation from intestinal stem cells in the epithelial crypts. While most specialized gut secretory cells, including goblet cells and Paneth cells, require the *Atoh1* transcription factor [18,19], its role in tuft cell formation is less well defined, with studies reporting both absent and increased tuft cells upon *Atoh1* deletion, with disparate results between embryonic versus adult animals, and between small and large intestinal tissues [20,21,22,23]. A likely explanation is that there are both *Atoh1*-dependent and -independent pathways that are initiated in a temporal- and tissue-specific manner [2].

As tuft cells mature, they also require a POU domain, class 2, transcription factor 3 (*Pou2f3*) [4], without which mice remain tuft-cell deficient; other characteristic genes in mice include Transient Receptor Potential Cation Channel Subfamily M Member 5 (*Trpm5*) [24], and Doublecortin-like kinase-1 (*Dclk1*). A suite of key genes are also expressed uniquely by tuft cells in the intestinal tract, such as *Gnat3* and *Gfi1b*, as well as *Il17rb* (encoding IL-25R), choline acetyltransferase (*Chat*), and arachidonic acid metabolism genes (*Cox1, Ptgds1, Alox5*) [11,20]. Tuft cells also express the succinate receptor (*Sucnr1*) as discussed below.

TRPM5 mediates a pivotal step in the taste signal transduction pathway [25], closely linking tuft cell function to taste sensation; tuft cells also express a set of G-protein coupled (GPCR) taste receptors, and, as detailed below, a loss of TRPM5 can ablate function in vivo [6] as is also found with the loss of certain taste receptor proteins such as Tas1R3 and Tas2r. However, taste receptors and TRPM5 are expressed by other cell subsets. In mice, enteroendocrine cells also express functional TRPM5, meaning that this product is not a specific marker for tuft cells [25,26].

Tuft cells exhibit significant heterogeneity, as highlighted by a single-cell RNA (scRNA) sequencing (scRNA-Seq) analysis [11] and antibody probing by immunofluorescence [27]. In the mouse small intestinal epithelium, two distinct populations of mature tuft cells, designated as tuft-1 and tuft-2, were distinguished by scRNA analysis. Both clusters exhibited Dclk1 expression; however, the tuft-2 cluster showed enrichment in immune-related genes, including Ptprc, which encodes the pan-immune marker CD45 [11]. This unexpected finding was confirmed using single-molecule FISH, revealing coexpression of *Dclk1* and *Ptprc* mRNA in certain tuft cells. Furthermore, the tuft-1 cluster displayed an enrichment of neuronal genes, suggesting that these cells mediate nervous system interactions as found in airway tuft cell populations [27,28]. In contrast, significant levels of the type 2-promoting cytokine TSLP were exclusively expressed in tuft-2 cells [11]. In addition, a novel subset of tuft cells showed immunoreactivity for 5HT (serotonin) localized to their apical surface, although not expressing the tryptophan hydroxylase (TPH) which is considered the enzyme required for 5HT synthesis [29].

As previously mentioned, the intestinal epithelium plays a crucial role in initiating and executing immune responses regulated by immune-specific cytokines such as IFNγ, IL-13, and IL-22 [4,5,6]. Notably, IL-13 induces BMP signalling, which functions as a negative feedback loop in limiting tuft cell hyperplasia driven by immune type 2 responses [30]. This feedback loop involves the inhibition of SOX4 expression to regulate the tuft cell progenitor population. Moreover, blocking BMP signalling with the ALK2 inhibitor DMH1 disrupts the feedback loop and increases tuft cell numbers in both in vitro and in vivo settings [30]. Overall, these novel insights into cytokine effector responses highlight the unexpected and crucial role of BMP signalling in type 2 immunity, offering potential opportunities for tailored epithelial immune responses. 

## 3. Tuft Cell Responses to Parasite Infection

The critical functional role of tuft cells was not appreciated until two studies in 2016 demonstrated that they sense and respond to intestinal helminth infection through the release of IL-25 which primes protective type 2 immune responses in the gut [4,5]. IL-25 acts as an alarmin to activate type 2 innate lymphoid cells (ILC2s) to produce IL-13; this key cytokine induces epithelial stem cells to differentiate into tuft cells (as well as goblet cells) in a positive feedback loop. The infection of mice with parasites that elicit type 2 immune responses, such as the intestinal nematode *Nippostrongylus brasiliensis,* drives tuft cell hyperplasia [4,5] concomitant with increased *Sox4* expression [21]. Significantly, *Pou2f3*-deficient mice were unable to expel the *N. brasiliensis* infection [4], as were *Sox4* deficient mice [21]. 

In addition, infections with an enteric protozoan (*Tritrichomonas muris*) can induce the expansion of tuft cells and corresponding activation of the type 2 response [6]. The activation of tuft cells during *Trit. muris* infection is known to be via the SucnR1 receptor [31,32] and also requires TRPM5 [33]. Another parasitic helminth, *Trichinella spiralis,* activates a Tas2R-mediated signalling pathway in intestinal tuft cells [34], while, in the case of *Trit. muris infection*, Tas1R3 regulates small intestinal tuft cell homeostasis as well as the Sucnr1 [6]. 

Downstream of tuft cell differentiation and expansion, there is an enhanced production of acetylcholine and arachidonic acid metabolites such as leukotrienes and prostaglandins [9]. Leukotrienes synergise with IL-25 in the activation of ILC2 cells, thereby contributing to anti-helminth immunity [35]. Notably, both acetylcholine and prostaglandins are also produced by airway tuft cells [36,37], indicating a general role in inflammatory responses to exogenous threats.

However, observations of tuft cell expansion are not always reproduced across other parasitic infections. The helminth *Heligmosomoides polygyrus* establishes a chronic intestinal infection in mice, accompanied by a relatively weak tuft cell response [38]. Notably, infection with *H. polygyrus* renders mice less responsive to *N. brasiliensis* with a subdued level of tuft cell activation. As detailed below, this has been attributed to secretory proteins released by *H. polygyrus* that target the intestinal stem cell differentiation programme.

## 4. Tuft Cells in Parasite Infections of Non-Murine Hosts

Tuft cells have been identified across a range of mammalian species from mice and ruminants to pigs and humans, generally under steady-state conditions rather than in the setting of infection. However, a detailed study of the presence and gene expression profile of tuft cells in sheep, following nematode infections using immunohistochemistry and single-cell RNA-sequencing, was recently published [39]. Tuft cells were characterised in the ovine abomasum, the true stomach of ruminants, and a significant increase in their numbers was observed after infection with the globally important nematodes *Teladorsagia circumcincta* and *Haemonchus contortus*. These ovine tuft cells have an enrichment of classical tuft cell gene markers such as *Pou2F3*, *Gfi1B*, and *Trpm5*, as well as genes associated with signalling and inflammatory pathways. Interestingly, it was also found that while murine tuft cells express the succinate receptor SucnR1 and free fatty acid receptor Ffar3 as “sensors”, these receptors were not found to be expressed in ovine tuft cells. Instead, an enrichment of taste receptor Tas2R16 and mechanosensory receptor Adgrg6 in ovine tuft cells was observed [39]. With progress in gastrointestinal organoid cultures from ruminant species, showing differentiation of the specialized cell types, further advances in defining these pathways are to be expected [40,41]. 

While less well understood in terms of their gene expression profile, tuft cells have also been identified in pig intestines as DCLK1^+^ epithelial cells, and have been shown to increase in number following infection with the porcine whipworm, *Trichuris muris* [42]. Interestingly, this study demonstrated that supplementation with inulin, a fructan which is fermented by commensal microbiota to produce short-chain fatty acids, synergistically enhanced intestinal Th2 and mucosal barrier gene expression, including *Dclk-1*, and elevated tuft cell numbers in whipworm infected pigs. It is unclear whether the immunomodulatory effects of inulin were due to direct effects on the host or were driven by the microbiota which were also altered in inulin supplemented pigs. Specifically, this demonstrates that dietary supplementation may be a practical way to modulate tuft cell numbers in livestock to enhance intestinal immune responses; more generally, it is interesting to speculate whether the extensive glycosylation present on many helminth secreted proteins [43] is detected by tuft cell taste receptors.

Much of the current information on tuft cells inevitably derives from murine studies, but, at least in the steady state, human intestinal tuft cells show similar patterns of gene expression to those from other mammalian species [44,45] and so may be predicted to respond to parasite infections in a similar manner. However, in one study which has been recently published, patients with chronic infections with the parasitic flatworm *Schistosoma* did not show an increased number of tuft cells in the large intestine compared to uninfected controls, although an increase in mucus production by large intestinal goblet cells was manifest [46]. Although this observation could suggest a lower abundance of tuft cells in humans, it should be noted that schistosomiasis patients have long-standing infections during which parasite-mediated down-modulation of host immunity may have taken place.

## 5. Tuft Cell Responses to Microbial Infection

Although tuft cell activation is most strongly associated with parasite infections, there are also key interactions with certain bacterial organisms. Although tuft cells are not activated by pathogenic bacteria such as *Salmonella enterica* [11], they do sense other bacteria, such as *Bifidobacterium* species, in a succinate-dependent manner [47]. On the other hand, in the colon, tuft cells are highly sensitive to intestinal bacteria and respond to changes in the microbiome [48]. Antibiotic-mediated depletion of the gut microbiome can affect tuft cell populations, and the presence of bacteria can alter tuft cell gene expression and expansion [27]; equally, tuft cell activation through succinate can change the spectrum of antimicrobial peptide expression, resulting in a significant change in the microbiome [49].

In the context of viral infections, tuft cells can be directly infected by certain viruses, such as murine norovirus (MNV) [48] and murine rotavirus [50]. In the case of MNV, tuft cells are the primary target for infection, and the virus can exploit the immune-privileged niche of tuft cells to evade immune responses. Norovirus targets tuft cells via the CD300lf receptor, while also promoting expression of type 2 cytokines, which act to counteract the antiviral response and thereby amplify norovirus infection [48,51]. This scenario is supported by further work showing that an active *H. polygyrus* infection exacerbated West Nile Virus (WNV) pathology by promoting Type 2 responses in a tuft cell-dependent manner, while pathology was ameliorated in either IL4R- or tuft cell-deficient mice [52]. In the rotavirus setting, tuft cells also became directly infected but showed no numerical expansion, and, in contrast to events during helminth infection, down-regulated IL-25 and leukotriene production, indicating a conventional type 1 antiviral state may be induced [50]. 

Overall, tuft cells play diverse roles in the immune response to parasites, bacteria, and viruses in different tissues [51]. They are also pivotal in the regulation of the commensal microbiota; for example, through sensing of succinate levels in the intestinal tract, tuft cells control Paneth cell gene expression, and thereby the level of anti-microbial peptides (AMPs) that differentially control microbial taxa in the intestine [49,53]. Thus, there is a continuum of tuft cell activation and secretion phenotypes that vary depending on the type of micro-organism or parasite involved and their location within the gastrointestinal tract. It will be fascinating to further explore how tuft cells contribute to the regulation of bacterial microflora, and how these interactions are associated with allergy and autoimmunity, given the correlation between reduced helminth prevalence, dysregulated microbiomes, and increased incidence of inflammatory disorders [54].

## 6. Search for Tuft Cell Ligands 

Unusually, perhaps, the repertoire of tuft cell receptors is far better characterised than the range of complementary ligands that bind those receptors [55]. Since tuft cell activation is dependent on Trpm5 that interacts with G-protein coupled taste receptors, it is expected that the ligands will be small molecules akin to known substances such as the bitter-tasting compounds denatonium or salicin. In addition, a separate line of investigation identified succinate as a major stimulant of tuft cells [32,47,53,56], and detectable succinate is released both by the *T. muris* protists, and *N. brasiliensis* helminths, presumably as the result of heightened anaerobic metabolism. However, succinate activation is not sufficient to drive anti-helminth immunity, and tuft cell expansion can take place in parasite-infected mice lacking the succinate receptor gene *Sucnr1* [56].

To date, no helminth-specific ligand has been identified, although a component in *T. spiralis* extracts and ES material has been found to activate the Tas2r receptor in murine tuft cells [34]. Studies have demonstrated that tuft-2 cells exhibit a high expression of the vomeronasal receptor Vmn2r26, which is responsible for detecting the bacterial metabolite N-undecanoylglycine (N-C11G). This metabolite is produced by *Shigella* bacteria, susceptibility to which is increased in mice lacking the gene *Sh2d6*, which is restricted to tuft-2 cells. Activation of this receptor triggers the production of prostaglandin D2 (PGD2) by tuft-2 cells. This PGD2 production facilitates mucus secretion by goblet cells, playing a role in protecting against bacterial infections [57].

Known taste receptor ligands have also been investigated for their effects on tuft cells in vivo. Berberine, a component of curcumin, activates Tas2Rs giving rise to tuft cell expansion and IL-25 production in mice [58], with systemic metabolic impact as described below.

## 7. Tuft Cells in Organoid Models

The use of new in vitro models, such as small intestinal organoids, has allowed the analysis of signals involved in tuft cell expansion and stem cell reprogramming, in the presence of parasitic infections [59,60]. 

In 2022, studies showed that not all helminths are able to activate tuft cell amplification on the same scale, and, specifically, the natural murine helminth *H. polygyrus* is able to inhibit this expansion even when accompanied by an overall type 2 response [38]. Furthermore, inhibition of tuft cell expansion was observed not only by live infection, but also by soluble excretory–secretory (ES) products from worms cultivated in vitro. Indeed, there was a significant down regulation of gene sets related to other defence-related cell types in organoid cultures exposed to *H. polygyrus* ES (HpES), suggesting a broader impact of this nematode on the intestinal epithelium rather than the targeting of specific cell lineages. Additionally, the presence of HpES has noticeable effects on the morphology of organoids, leading to the formation of large spheroid structures with limited crypt budding. This highlighted two morphological organoid types: the immature “spheroid” and the mature “budding” morphology [30]. Hence, both at the level of gene expression and cellular organisation, this parasite redirects intestinal stem cell differentiation. A particular change induced by the parasite is the expansion of a specific subset of stem cells called revival stem cells, which are characterised by the expression of the molecular chaperone gene *Clusterin* [61]. These reserve stem cells exhibited enhanced self-renewal capacity and a reduced ability to differentiate into specialized cells involved in type 2 immunity. Taken together, these findings suggest that by suppressing tuft cell expansion, some parasites may promote their own survival, while also enhancing epithelial regeneration through stem cell modulation.

## 8. Tuft Cells and Metabolic Regulation

In recent years, increasing attention has been paid to the involvement of tuft cells in metabolic regulation and their impact on whole-body metabolism, as would be expected from their ability to sense various dietary components, such as fatty acids, sugars, and bile acids, as well as metabolic signals from the microbiota. In addition, tuft cells are involved in the communication between the gut and the brain, via the gut–brain axis, releasing signalling molecules that modulate the localised and central nervous systems, such as acetylcholine. The bidirectional communication between the gut and the brain is implicated in the regulation of food intake, energy expenditure, and metabolic homeostasis [24,31,62].

Imbalances in tuft cell-mediated immune responses have been linked to chronic inflammation and metabolic disorders. In conditions like obesity and high-fat-diet-induced metabolic syndrome, there can be changes in tuft cell gene expression, hormone secretion, and responsiveness to nutrients, contributing to metabolic dysregulation and its associated complications [31,56]. Mice exposed to a high-fat diet (HFD) showed increased levels of the metabolite succinate in the small intestine. Succinate acted to trigger tuft cells to produce the cytokine IL-25 which, in turn, activated the ILC2/IL-13 cascade for type 2 immune responses that may counteract metabolic dysfunction [31]. 

IL-25 appears to be a key mediator of metabolic function, as administration to obese mice on a high-fat diet leads to more moderate weight gain, reduced adipose tissue mass, and improved glucose metabolism, alongside increased numbers of ILC2s, type I and type II natural killer cells, eosinophils, and alternatively activated macrophages in the visceral adipose tissue [63]. Given that succinate can trigger type 2 immunity using the IL-25 released by tuft cells, it is tempting to hypothesize that succinate, whether dietary or microbiome-derived, could potentially yield similar metabolic benefits, such as improved glucose metabolism and decreased adipose tissue [56].

There have also been recent reports of tuft cell numbers changing under different high-fat diets or supplements, directly implicating them in regulating energy metabolism. Mice exposed to a high-fat diet showed decreased numbers of type 2 tuft cells expressing IL-25 and TSLP, with reductions correlating to greater weight gain [64]. Conversely, Chen et al. showed that indoleproprionic acid (a metabolite of tryptophan) promotes gut integrity, with a reduction in colonic inflammation and an expansion in tuft cells resulting in protection against obesity [65]. Finally, it has been reported that berberine, an extract widely used in traditional Chinese medicine to treat ulcerative colitis, interacts with the Tas2r receptor on tuft cells. Berberine has been proposed to participate in the activation of type 2 immune responses via tuft cells, which may underpin its effects in suppressing obesity [58,66].

The emerging evidence for tuft cell functions in metabolic regulation requires further investigation into whether they are primary instigators that drive downstream regulatory populations (such as alternatively activated macrophages) or act in a redundant fashion alongside other populations involved in maintaining metabolic homeostasis. In either case, modulating tuft cell function and signalling pathways could offer novel approaches to manage conditions such as obesity, diabetes, and metabolic syndrome.

## 9. Tuft Cells in Intestinal Homeostasis and Cancer

Tuft cells have been of particular interest in relation to intestinal cancer, both as tuft cell carcinoma (TCC) and malignancies of other epithelial cell types [67]. TCC is a rare and aggressive form of colorectal cancer that has been suggested to arise from the malignant transformation of tuft cells in the intestinal epithelium [23]. While tuft cells are a normal component of the intestinal lining, the exact mechanisms underlying their transformation into cancer cells are not fully understood. However, experimental studies have provided insights into potential signalling pathways and genetic alterations that may be involved [20]. 

In one example, Westphalen et al. proposed that mutated tuft cells can act as cancer-initiating cells in the context of a major tissue injury, showing that genetic alterations in tuft cells (particularly loss of the tumour suppressor gene *Apc*), in conjunction with the inflammatory stress of DSS colitis, can lead to tumour development [23]. Subsequent studies suggested that in the gut it is more likely that mutations acquired by stem cells or progenitors can be passed on to tuft cells, which then serve as cancer-initiating cells. These mutations may include critical genes such as *Ptgs2*, which is upregulated in colorectal cancer, or genes involved in cholinergic signalling directly modulating Wnt signalling [62]. Another study explored the normal functions of tuft cells and their potential involvement in colorectal cancer [20]. 

The identification of unique gene markers in tuft cells has enabled the exploration of their association with cancer [14]. For instance, DCLK1 has been found to be upregulated in renal clear cell cancer, pancreatic cancer, and colorectal cancer [68,69,70,71]. Moreover, KRAS mutant Dclk1^+^ cells can functionally contribute to the pathogenesis of pancreatic cancer and, therefore, are a potential target in cancer treatment [72]. Additionally, silencing DCLK1 has been shown to hinder the progression of pancreatic ductal adenocarcinoma and colon cancer, indicating its tumour-promoting role in cancer [73]. While the involvement of tuft cells in tumours will inevitably vary depending on the tumour type and stage, targeting tuft cells could potentially lead to novel immunotherapy strategies in cancer treatment.

## 10. Conclusions

From relative obscurity, tuft cells have rapidly come to the fore in understanding the physiology and immunology of barrier tissues, playing key roles in metabolic sensing and regulation, immunity to parasites, and pathways to cancer (Figure 1). While our focus has been on intestinal tuft cells, they act not only at barrier sites but also in the thymic epithelium at the site of T cell selection [74,75]; in most settings they are closely coupled with the nervous system as key sensors of infectious or noxious threats [76], able to drive the defences most appropriate to the tissue in question [2]. 

The unique transcriptional profiles of tuft cells in the intestinal tract, airways, and thymus remain to be fully understood; their receptor repertoire has yet to be matched with physiological, and pathogen-derived ligands, and the functional importance of pharmacologically active products such as leukotrienes and acetylcholine needs to be identified in both steady-state and infection settings. Taken together, new investigations into these areas are likely to be scientifically illuminating, paving the way for novel therapeutic approaches to modulate tuft cell responses in diseases and infections in the intestinal tract and other tissues.

## Figures and Tables

**Figure 1 cells-12-02477-f001:**
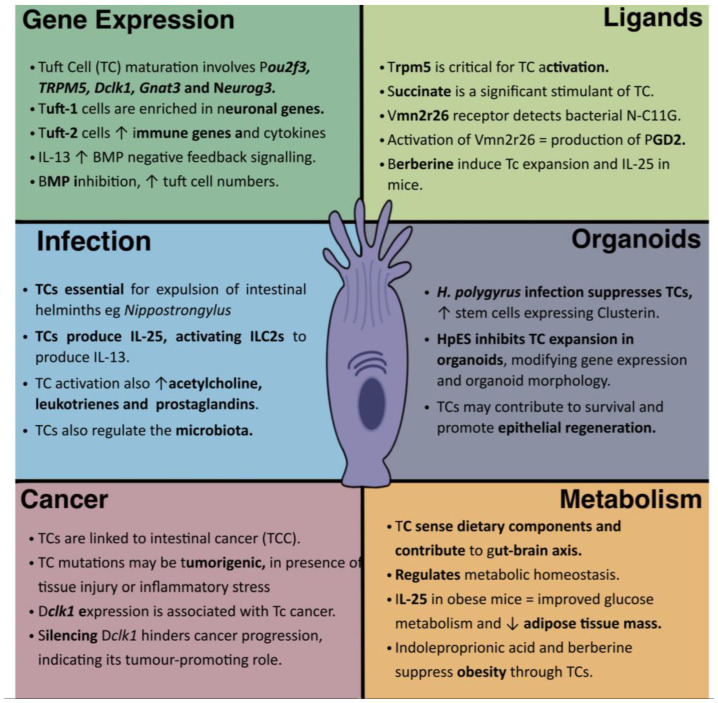
Tuft Cell gene expression, functions and implications in disease. This figure illustrates the diverse characteristics and roles of tuft cells (TCs) in various homeostatic and pathological contexts, including gene expression, and roles in infection and cancer, together with their ligands identified to date and potential role in metabolic regulation.

## Data Availability

Not applicable.

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
