# Peer review of "Tuft Cells: Detectors, Amplifiers, Effectors and Targets in Parasite Infection"

_cells, 2023, doi:10.3390/cells12202477_

Round 1

Reviewer 1 Report

This is a well-written manuscript that thoroughly covers a number of original and recent studies about tuft cell functions in the intestine. The authors statement for each section is scientifically sound based on their background of immunology. There are only few unclear sentences and this reviewer requests minor modifications.

1.     This article cites 11 review papers, which is too many for a review paper. Please consider to reduce review citation.

2.     Is Reference 10 still under resubmission? Papers must be cited at lease after official acceptance.

3.     This paper underestimates enteroendocrine cells (EEC) as sensory cells utilizing TRPM5. Taste receptors and TRPM5 are expressed in a subset of tuft cells and considerable number of enteroendocrine cells have functional TRPM5 in mice (Ref. 25 and PMID: 17030556). It has been an argument how to distinguish nutrient sensory function mediated by TRPM5 between EEC and tuft cells, since TRPM5 is not solely tuft cell marker.  Please add a sentence about it.

4.     Line 98-99: Reference 30 describes 5-HT immunoreactivity in some tuft cells but no evidence for 5-HT production or expression (5HT is not a gene or protein). The #30 paper clearly states that HTP is not colocalize with DCLK1. Please rewrite this sentence.

5.     Line 126-127: Reference #6 looks wrong. It should be Howitt et al. PMID 31980480?

6.     Line 131: Please show evidence for luminal secretion of Ach from tuft cell.

7.     Line 216: Reference 51 describes “revival stem cell” but this manuscript uses “reserve stem cell.” Is there particular reason why terminology was changed? Please explain.

8.     Line 229: In my knowledge, there is no clear evidence for bidirectional communication between tuft cells and the brain. If there is any, please add correct reference and a statement. It is related to comment #3.

9.     Figure 1 is not cited in main text in manuscript I have received.

10.  In the Figure, BMP inhibition must be in Ligand category, and Trpm5 is in Gene category: BMP is a secretory ligand modulating cell differentiation. Trpm5 function was only studies by using Trpm5 gene knockout mice.

Reviewer 2 Report

The manuscript (cells-2603190) entitled "Tuft Cells: Detectors, Amplifiers, Effectors and Targets in Parasite Infection " This is a very interesting review article that raises the important question of the role of Tuft cells in understanding the physiology and immunology of barrier tissues and plays a key role in regulating metabolism, response to infection and cancer pathways. The article points the way to new therapeutic approaches for modulating tuft cell responses in diseases of the gastrointestinal tract and other tissues.

Comments:

The work has a solid scientific basis and is an important inspiring topic for other new scientific research and deserves publication. 

Reviewer 3 Report

This is a thorough and well written generalist's review of Tuft cells with a focus on parasite infections. As a practitioner at the very applied end of parasite immunology I enjoy reading this type of review paper. It allows learning and understanding without getting too consumed in the fine detail.

Some comments.

Given that the paper is primarily around Tuft cells and parasite infections, section 9 (Tuft Cells in Non-Murine Hosts) I feel should follow section 3 (Tuft Cell responses to parasite infection).

Line 65 states “In the healthy, steady-state intestinal tract, tuft cells are found at a low frequency”. As most mammals are not housed in a clean cage but instead consistently exposed to parasite challenge e.g., livestock grazing pasture, what really is “steady-state”? What are tuft cell numbers/turn-over in animals with a fully developed protective immune response to parasites? One could argue that normal healthy steady-state is an animal able to handle/repel constant parasite challenge. Perhaps the authors could better describe this statement?  

Line 157 Is WNV supposed to be MNV, otherwise explain what is WNV?

 Line 172. I wonder if "reduced helminth burdens" is not totally correct and perhaps "reduced helminth exposure" is a better descriptor here.

Line 201 Section 6 could do with a reference e.g., White, R., et al. (2022). "Organoids as tools to investigate gastrointestinal nematode development and host interactions." Frontiers in Cellular and Infection Microbiology 12.

Line 222. In light of the discussion in Section 7 (Tuft cells and Metabolic Regulation), could the authors make comment on the inappetence induced following gastrointestinal parasite infection and any possible role of Tuft cells in this process? 

Line 296. Section 9 could do with a few more references after the first sentence, Section 9 gives emphasis on the authors own research, and summarizes this work (ref 64), however as a review paper it definitely needs to quote other previous papers on the subject matter.  e.g., Faber, M. N., et al. (2022). "Development of Bovine Gastric Organoids as a Novel In Vitro Model to Study Host-Parasite Interactions in Gastrointestinal Nematode Infections." Frontiers in Cellular and Infection Microbiology 12.

Line 309. Incorrect spelling of been (bene).

The use of acronyms in such a review paper could be helped by having a box adding a description for the less common ones. While many acronyms are co-described in the manuscript some need better descriptions e.g., line 97, TSLP; line 102, BMP; line 104 Sox4, line 154 CD300lf.

There is conflict between the generalized statement that succinate could be used expand tuft cell activity (line 40-42) and the negative feedback loop described in lines 102-103 that limit tuft cell hyperplasia. Some explanation is required.      

A thought: the authors quote the impact of Inulin which is made up of Fructose as a possible stimulus of elevate Tuft Cells. Glycans of nematodes (N- O- and lipo-linked) are quite abundant and in some cases have been shown to be directly involved in immune protection but seem to be ignored in research studies. Are there “sweet” sensors on tuft cells?  
